# UAV Autonomous Tracking and Landing Based on Deep Reinforcement Learning Strategy

**DOI:** 10.3390/s20195630

**Published:** 2020-10-01

**Authors:** Jingyi Xie, Xiaodong Peng, Haijiao Wang, Wenlong Niu, Xiao Zheng

**Affiliations:** 1Key Laboratory of Electronics and Information Technology for Space System, National Space Science Center, Chinese Academy of Sciences, Beijing 100190, China; xiejingyi18@mails.ucas.ac.cn (J.X.); niuwenlong@nssc.ac.cn (W.N.); zhengxiao@nssc.ac.cn (X.Z.); 2University of Chinese Academy of Sciences, Beijing 100049, China; 3Alibaba Damo Academy, Hangzhou 311121, China; wanghaijiaocool@126.com

**Keywords:** quadrotor unmanned aerial vehicle, deep reinforcement learning, autonomous tracking and landing

## Abstract

Unmanned aerial vehicle (UAV) autonomous tracking and landing is playing an increasingly important role in military and civil applications. In particular, machine learning has been successfully introduced to robotics-related tasks. A novel UAV autonomous tracking and landing approach based on a deep reinforcement learning strategy is presented in this paper, with the aim of dealing with the UAV motion control problem in an unpredictable and harsh environment. Instead of building a prior model and inferring the landing actions based on heuristic rules, a model-free method based on a partially observable Markov decision process (POMDP) is proposed. In the POMDP model, the UAV automatically learns the landing maneuver by an end-to-end neural network, which combines the Deep Deterministic Policy Gradients (DDPG) algorithm and heuristic rules. A Modular Open Robots Simulation Engine (MORSE)-based reinforcement learning framework is designed and validated with a continuous UAV tracking and landing task on a randomly moving platform in high sensor noise and intermittent measurements. The simulation results show that when the moving platform is moving in different trajectories, the average landing success rate of the proposed algorithm is about 10% higher than that of the Proportional-Integral-Derivative (PID) method. As an indirect result, a state-of-the-art deep reinforcement learning-based UAV control method is validated, where the UAV can learn the optimal strategy of a continuously autonomous landing and perform properly in a simulation environment.

## 1. Introduction

In recent years, with the rapid development of unmanned aerial vehicle (UAV) technology, the UAV has been widely used in military and civilian fields, such as search, rescue, exploration, and surveillance [1,2]. Autonomous tracking and landing is a key point in UAV application [3,4,5,6]. However, compared to landing in a simulated environment or a static platform, autonomous tracking and landing is more difficult because classical techniques have their limitations, in terms of model design and non-linearity approximation [7]. Furthermore, due to the absence of precise sensors and the constraint of the sensors’ specific physical motion, autonomous tracking and landing works poorly with high sensor noise [6,8] and intermittent measurements [9,10].

Given the importance and complexity of the UAV tracking and autonomous landing, increasingly more scholars from different fields have shown interest in specific solutions such as perception and relative pose estimation [11,12] or trajectory optimization and control [13,14]. Regarding the control maneuvers when the relative state of the moving platform is assumed to be known, the Proportional-Integral-Derivative (PID) controller is the mainstream algorithm for aggressive landing from relatively short distances [4,5,15], but a fixed gain of the PID controller cannot provide immediate response to overcome a nonlinear effect, moreover, PID gain tuning is a crucial part and needs a lot of effort for optimal gain [16]. This calls for nonlinear control approaches for more precise control of UAV. By applying a state-dependent coefficient (SDC)-based nonlinear model inversion, the authors of reference [17] eliminated the need for linearization of the aircraft dynamics, but this approach is quite sensitive towards sensor noise. In order to solve the problem of UAV autonomouslanding under special circumstances, such as landing on top of a moving inclined platform [14,18], or landing on a platform moving in a figure of eight [3,19] (the task of the competition in Mohamed Bin Zayed International Robotics Challenge (MBZIRC) 2017), a Model Predictive Controller (MPC) tracker was used to generate UAV feasible trajectories, which could minimize an error of UAV future states over a prediction horizon to fly precisely above the car given the dynamical constraints of the aircraft. MPC holds the ability to anticipate upcoming events and can yield control inputs accordingly, however, the development of accurate prediction models requires a tiresome design effort. Sliding mode methods have been widely used in UAV autonomous tracking and landing control algorithms [20,21,22]. This approach changes the UAV nonlinear dynamics by the application of a discontinuous control signal, but the main issue with sliding mode control is chattering and a high control demand [7]. Unlike the linear and non-linear controllers mentioned above, a fuzzy logic controller does not depend on a precise mathematical model. Regardless of the good performance by PID controllers, these still need to be adaptive for uncertain conditions. To achieve this purpose, PID control was extended further to fuzzy adaptive PID [23,24], and the authors of reference [16,25] presented a fuzzy logic-based UAV tracking and landing using computer vision.

Different from the above methods in solving the control of a UAV by building a prior model and then making a decision based on a dynamics model, reinforcement learning is an ideal solution to deal with unknown system dynamics in different tracking and landing circumstances [26]. Recently, significant progress has been made by combining deep learning with reinforcement learning, resulting in the deep Q-learning network (DQN) [27], the deterministic policy gradient (DPG) [28], and the asynchronous advantage actor-critic (A3C) [29]. These algorithms have achieved unprecedented success in challenging domains, such as the Atari 2600 [27].

Concerning deep reinforcement learning for UAV autonomous tracking and landing tasks, a hierarchy of DQN was proposed in [30,31], which is used as a high-end control policy for the navigation in different phases. However, the flexibility of the method is limited because the UAV’s action space is defined by discrete space rather than continuous space. Furthermore, the authors of reference [32] addressed the full problem with continuous state and actions spaces based on Deep Deterministic Policy Gradients (DDPG) [33]. However, the altitude (*z*-axis) is not included in the framework, there is a significant design effort during the reward plasticity, and the proposed Gazebo-based reinforcement learning framework has a weak generalization capability, which results in less autonomous agents.

In general, the current research on the problem of a UAV autonomous tracking and landing is mainly focused on the situation in which the platform has a constant speed. However, more sophisticated control is required to operate in unpredictable and harsh environments, such as high sensor noise and intermittent measurement. In terms of the trajectory control or classic controller, e.g., PID, nonlinear, fuzzy logic controllers are the mainstream algorithm, but these techniques are limited to model design, non-linarites approximation, and disturbances rejection. Although, in several studies, the deep reinforcement learning (DRL)-controller has performed well in UAV tracking and landing problems, the control problem of UAV tracking and landing on a randomly moving platform becomes intractable, due to the low reliability and weak generalization ability of the mathematical model.

We decompose the problem of landing on a moving platform into two aspects: one is perception and relative pose estimation, the other is trajectory optimization and control. The reader is referred to [6,34] for algorithms which are used particularly for landing on a platform using visual inputs from UAV. In the present paper, we focus on trajectory planning and control, aiming to solve the control problem of UAV tracking and landing on a moving platform. We build a novel model-free-based tracking and landing algorithm to solve the problem of sensor noise and intermittent measurements. First, by taking the sensor noise, intermittent measurements, and randomness of UAV movement into consideration, a novel dynamic model based on the partially observable Markov decision process (POMDP) [35] is built to describe the autonomous process of UAV tracking and landing. Then, an end-to-end neural network is used to approximate the action controller of UAV autonomous tracking and landing. Finally, a DRL-based algorithm is adopted to train the neural network to learn from the tracking and landing experience.

The rest of this article is organized as follows. In Section 2, we introduce a UAV autonomous tracking and landing model based on the POMDP. In Section 3, we discuss the hybrid strategy included with deep reinforcement learning and heuristic rules in order to calculate the optimal control output and realize UAV autonomous tracking and landing tasks. In Section 4, we present the experimental results of our methods on a Modular Open Robots Simulation Engine (MORSE) simulator. Conclusions and future work are drawn in Section 5.

## 2. Problem Definition

Generally, UAV autonomous tracking and landing could be described as follows: Given a moving platform, such as a car or truck, and a UAV, the moving state of the UAV and platform could be detected by some kind of sensors; however, only part of the moving state of the platform could be observed. The observation states of both platform and UAV may have some errors and information update delay, so the observation could not be regarded as accurate. The task of UAV autonomous tracking and landing is to control the UAV with a proper speed or accurate speed, so as to land on the platform properly. Unlike previous research [30,32,36], in this paper, incomplete and inaccurate observations are taken into consideration, which makes it more consistent with real situations.

### 2.1. UAV Autonomous Landing System

As mentioned above, an autonomous tracking and landing system consists of a UAV and platform (shown in Figure 1), and, to keep the example simple, we assume that the sensors attached to the UAV and platform could only observe the speed and position of the UAV and platform. In addition, the observations of both the UAV and platform have a certain frequency, so no real-time information could be observed, such as the state update delay, making autonomous tracking and landing more difficult.

Naturally, observations of a system have a certain frequency, which means that multiple states with a timeline could be obtained, as shown in Figure 2. At each time a state is observed, a proper decision should be made (which usually means some control parameters, such as speed), and then a new state could be observed and a new decision should be made. Such a process would continue until the UAV lands successfully or reaches a certain step state. Therefore, the control of UAV autonomous tracking and landing can be regarded as a sequential decision problem.

### 2.2. POMDP Mathematical Model

As mentioned above, the UAV autonomous tracking and landing process is partially observable and can be seen as a sequential decision problem. Therefore, the partially observable Markov decision process is adopted to describe this process.

Typically, the POMDP consists of tuples (S,A,T,R,O,Ω,γ), where *S* is a set of states, *A* is a set of actions, and *Ω* is a set of observations. *O* is a conditional observation probability and the functions *T* and *R* and factor *γ* define a Markov decision process [37], where *T* is a transition function, *R* denotes immediate rewards, and *γ* is a discount factor. Based on the POMDP mathematical model, the UAV autonomous tracking and landing model is described as follows:
State *S*: Represents the state of the system, including the state of the UAV and moving platform; the system coordinates are constructed as in Figure 3. On the basis of this coordinate system, the state information could be described by the respective speeds and positions of the UAV and platform, as in Equation (1):(1)$$s\;=\;\left\{X_{u},Y_{u},Z_{u},v_{ux},v_{uy},v_{uz},X_{t},Y_{t},Z_{t},v_{tx},v_{ty},v_{tz}\right\}$$
where $v_{ux},v_{uy}\in \left[-10,10\right],\;v_{uz}\in \left[-1,3\right],\;v_{tx},v_{ty}\in \left[-5,5\right],\;v_{tz}\;=\;0$.Action *A*: The speeds of the UAV are used as action parameters defined as $A=\left\{v_{x},v_{y},v_{z}\right\}$.Transition function *T*: Represents the dynamic model of the UAV autonomous tracking and landing system, which is difficult to model and describe explicitly, in this work the transition model is not given (model-free).Observation *Ω*: Owing to the physical limitations of the sensor, one could only obtain the velocity and position information of the UAV and moving platform with sensor noise or intermittent measurements, and the observation is defined as
(2)$$\Omega \;=\;\left\{{X_{u}}^{\prime },{Y_{u}}^{\prime },{Z_{u}}^{\prime },{v_{ux}}^{\prime },{v_{uy}}^{\prime },{v_{uz}}^{\prime },{X_{t}}^{\prime },{Y_{t}}^{\prime },{Z_{t}}^{\prime },{v_{tx}}^{\prime },{v_{ty}}^{\prime },{v_{tz}}^{\prime }\right\}.$$Reward function *R*: Because one of our goals is to minimize the distance between the UAV and the moving platform, a positive reward of 10 is given when the distance between the UAV and the platform is less than a certain threshold, and a negative reward of −10 is given if the distance between the UAV and the platform is more than a certain threshold.

Additionally, due to the complexity of the UAV control, when the agent explores its environment in the early stages, the event of randomly reaching the distance between the UAV and the moving platform within a certain threshold is rare, and thus cannot provide the agent with enough information to converge. Conversely, it is the best to receive reward signals at each time step of UAV tracking, so the intermediate reward between a positive and negative reward is designed, as shown in Equation (3), where the agent maximizes the reward (thereby minimizing distance) in order to track the target as accurately as possible.
(3)$$R\;=\;\{\begin{array}{c} -10,dist>6\\ -0.1\times dist,0.8\leq dist\leq 6\\ +10,dist<0.8 \end{array}$$
where the variable dist is given by
(4)$$dist\;=\;{\sqrt{{\left(X_{u}\;-\;Y_{t}\right)}^{2}\;+\;{\left(X_{u}\;-\;Y_{t}\right)}^{2}}}^{}.$$

6.Discount factor *γ* is a parameter used to incorporate the future reward into current action. We used an action-value function to evaluate the learning outcome, which describes the expected accumulated discounted reward after taking an action $a_{i}$ in state $s_{i}$ and, thereafter, following policy *π*:(5)$$Q^{\pi }(s_{i},a_{i})\;=\;R\left(s_{i},a_{i}\right)\;+\;\gamma \sum_{s_{i+1}\in S}P_{s_{i}s_{i+1}}^{a_{i}}v^{\pi }(s_{i+1})$$
where $S_{i+1}$ denotes the state information on the next moment, $P_{S_{i}S_{i+1}}^{a_{i}}$ is the Markov dynamics model of the system, and $v^{\pi }$ is the total expected reward under the strategy π. The goal of this paper is to find the strategy $\pi (a_{i}|s_{i})$ to maximize $Q^{\pi }\left(S_{i},a_{i}\right)$, which means finding an optimal strategy for selecting actions based on the information observed by the UAV to realize UAV autonomous tracking and landing tasks.

## 3. UAV Autonomous Tracking and Landing Method Based on Hybrid Strategy

If the UAV autonomous landing problem can be described by an accurate mathematical model, then we can solve the objective function of the Markov decision model (shown in Equation (5)) by an iterative solution based on the direct method or indirect method of the optimal control theory and then directly obtain the above optimal decision strategy of the Markov decision process. However, as described in Section 2, the model of the target system is unknown and difficult to describe. Much of the current research [38,39] uses a reduced model of the UAV for generating the optimal landing trajectory for landing on a moving target. However, such an approximation may not be accurate, and, consequently, the landing performance is decreased. Besides, most methods take a state as inputs and assume that the observation of the state is accurate, which is unlikely to be true. Different from the previous definition of optimal control [40], the UAV optimal control method we propose refers to the control strategy that maximizes the action value function (shown in Equation (5)) of the UAV. Aided by reinforcement learning, a model-free method is adopted, and a novel network is used to map the observation to optimal action in an end-to-end way. In addition, we use the tracking and landing experience history to train the network to learn the optimal tracking strategy. Once the platform is tracked, a rule-based landing strategy is used to land the UAV.

### 3.1. Hybrid Strategy Method

The hybrid strategy adopted in this paper is shown in Figure 4. The strategy consists of two parts: tracking and landing modules. The tracking module introduces the reinforcement learning method to adjust the speed of the UAV in the horizontal direction, aiming to achieve the stable tracking of the moving platform. The landing module adjusts the height of the UAV in the vertical direction based on heuristic rules, so as to land the UAV on the platform.

Tracking module: The UAV autonomous tracking and landing problem has a continuous state and decision space, and the DDPG, which combines the DQN and DPG, is a deterministic strategy for a continuous action space. This method combines reinforcement learning with deep learning and has good potential for dealing with complex tasks. Thus, this network is introduced in this paper for mapping the observation to proper action in an end-to-end way.

Details of the decision network structure are shown in Figure 4. The network adopts the actor-critic architecture [41], in which the actor network input is the system state, mainly including the motion state information of the UAV and moving platform in the system. The output layer is a two-dimensional continuous action space, which corresponds to the speed value of the UAV in the longitudinal and lateral directions after scale conversion. The critic network estimates the action-value function that describes the expected reward after following policy *π*. Since the decision is a deterministic action, to ensure that the environment is fully explored during the training process, we constructed a random action by adding noise sampled from a noise process *N_i_*, in which the noise is only needed in the training process:(6)$$a_{i}\;=\;\mu \left(s_{i}|\theta^{\mu }\right)\;+\;N_{i}$$
where $a_{i}$ is the output action, $s_{i}$ represents the current state information, $\mu (s_{i}|\theta^{\mu }$) represents the decision taken when the policy parameter is $\theta^{\mu }$ in state $s_{i}$, and $N_{i}$ is the artificially added Gaussian noise attenuated over time.

As shown in Figure 4, the network consists of three fully connected layers, the FC1 and FC3 layers are followed with the relu activation function, and the FC2 layer is followed with the tanh layer. The parameters of each layer in the network are shown in Table 1. Furthermore, the effects of different network parameters on the UAV tracking performance are shown in Appendix C.

Landing module: The landing module adjusts the height of the UAV in the vertical direction based on heuristic rules (shown in Table 1). As show in Table 2, *dist* has been defined in Equation (4), and *height* is defined as $height=\;Z_{u}-Z_{t}$. According to the rules table, the speed of the UAV in the vertical direction depends on the distance and height between the UAV and the moving platform. When the distance between the UAV and moving platform is less than 4 m, the UAV should gradually reduce its height while ensuring stable target tracking. When the relative height between the UAV height and moving platform is less than 0.1 m, and the distance error of the horizontal direction is less than 0.8 m, it is then considered that the landing task is successful. When the target is lost during landing, the UAV would stop landing and gradually restore the initial height and re-plan the landing trajectory.

### 3.2. Network Model Training

To train the UAV tracking neural network to learn the optimal tracking strategy, we adopted the reinforcement learning process (shown in Figure 5). At each step, the UAV observes the states’ partially observable information and then interacts with the environment through actions while receiving immediate reward signals. After multi-step decisions, the agent gains decision-making experience, so as to obtain more cumulative rewards and maximize the action-value function (defined in Section 2).

One challenge when training a neural network is that there is a correlation between the data generated by sequential exploration in the UAV autonomous tracking and landing environment. To address these issues, a replay buffer is used to define a control experience tuple: *D_i_* = {*s_i_*, *a_i_*, *r_i_*, *s_i+1_*}, indicating the UAV input state at time *i*, outputting the control action, receiving the reward, and obtaining the state at the next time *i* + 1. The tuple is stored in the replay buffer, and the neural network is updated by uniform random sampling of the mini-batch data in the replay buffer.

The decision network training method is now presented. The critic network uses the method of minimizing the loss function to approximate the value function, which is defined as
(7)$$L\left(\theta^{Q}\right)\;=\;\frac{1}{N}\sum_{i}{\left[Q(s_{i},a_{i}|\theta^{Q})\;-\;r_{i}\;-\;\gamma Q^{\prime }\left(s_{i+1},\mu^{\prime }\left(s_{i+1}|\theta^{\mu^{\prime }}\right)\right)|\theta^{Q^{\prime }}\right]}^{2}$$
where $L\left(\theta^{Q}\right)$ is the loss function, $Q\left(s_{i},a_{i}|\theta^{Q}\right)$ is the estimate of the *Q* value at time *i* (the latter two values are the actual *Q* values after the action $a_{i}$ at time *i*), γ is the discount factor, and $r_{i}$ is an immediate reward.

In the actor network, the neural network is also used to approximate the strategy function, and the actor policy is updated to output the optimal decision on the basis of the current state. The updated formula is
(8)$$\nabla J\;\approx \;\frac{1}{N}\sum_{i}\nabla_{a}Q\left(s,a|\theta^{Q}\right){|}_{s=s_{i},a=a_{i}}\nabla_{\theta^{\mu }}\mu \left(s|\theta^{u}\right){|}_{s=s_{i}}$$
where ∇J represents the gradient direction of the *Q* value caused by the strategy μ, thereby updating the policy parameter $\mu \left(s|\theta^{u}\right)$, $\nabla_{a}Q\left(s,a|\theta^{Q}\right){|}_{s=s_{i},a=a_{i}}$ represents the change in the *Q* value generated by the action $a_{i}$ in the current state, and $\nabla_{\theta^{\mu }}\mu \left(s|\theta^{u}\right)$ represents the current policy gradient direction.

To ensure the stability of the learning process, in this paper, the networks $\mu^{\prime }\left(s|\theta^{u^{\prime }}\right)$ and $Q^{\prime }\left(s,a|\theta^{Q^{\prime }}\right)$ are created to be the same as the actor and critic network, respectively, which are then used for calculating the target values, and the parameter-updated formula is
(9)$$\begin{array}{ll} \theta_{i+1}{}^{Q^{\prime }}\; & =\;\tau \theta^{Q}\;+\;(1\;-\;\tau )\theta_{i}{}^{Q^{\prime }}\\ \theta_{i+1}{}^{\mu^{\prime }}\; & =\;\tau \theta^{\mu }\;+\;(1\;-\;\tau )\theta_{i}{}^{\mu^{\prime }} \end{array}$$

## 4. Simulation Results

### 4.1. Simulation Environment

To evaluate the behavior of the control system approach, experiments were performed on the Modular Open Robots Simulation Engine (MORSE) simulation platform (https://github.com/morse-simulator/morse), which can perform accurate dynamic simulations based on the state-of-the art Bullet library. The simulation environment was established using a UAV and ground vehicle. The UAV speed was controlled by the algorithm proposed in this paper, and the orientation of the ground vehicle (robot) was able to be changed at any time from the command line. The experimental computer was an AMD Ryzen 71,700 (eight-core processor, main frequency 3.0 GHz, 8 GB DDR4, 2400 MHz memory, operating system Ubuntu 16.04) running Python v3.6.0, anaconda v4.2.0, and TensorFlow v1.4.0.

The flowchart of the simulation experiment for the proposed method is shown in Figure 6. The flowchart defines two different types of tests. The first was to test the performance of the UAV tracking model by tracking moving platform with a fixed height of 5 m. Once the tracking error was less than a certain threshold, a second experiment could be carried out to evaluate the ability of the hybrid strategy method to automatically land from a height of 5 m. Both tracking and landing tests have the same environment: The tracking and landing phases were trained in simulation throughout approximately 60 k training steps. The trajectory of the moving platform could be linear, circular, or random. The maximum velocity of UAV and moving platform were defined in Section 2. The permitted horizontal area for the moving platform to move is a rectangle of 37 m × 85 m. The velocity controller frequency of UAV is 20 Hz in a real-time MORSE simulation.

In the simulation environment, in order to ensure the flight safety of the UAV, on the one hand, during the model training phase, we restricted the speed data output by the model, and stipulated that the horizontal velocity of the UAV $v_{x},v_{y}$ was less than 10 m/s, and $v_{z}$ was less than 3 m/s. On the other hand, we introduced heuristic rules to prevent the UAVs from crashing and keep them at a safe flying height, specifically, when the height of the drone was less than 3.5 m, $v_{z}$ = 0.5 × *height*. When the distance between the UAV and the moving platform was greater than 6 m, the flying height of the UAV must gradually return to the initial height of 5 m.

### 4.2. Autonomous Tracking Tests

In the autonomous tracking tests, the UAV and ground vehicle were randomly placed at an initial position. The behavior of the control system was evaluated using the root-mean-square error (RMSE) and tracking success rate (TSR) during the tracking tests. In Equation (10), the tracking success is defined as the horizontal distance between the UAV and moving platform, which is less than 3 m, and the definition of TSR is shown as follows:(10)$$TSR\;=\;\overset{N}{\underset{i=0}{\sum }}\frac{D_{i}}{N}\;\times 100\%,\;D_{i}\;=\;\left\{\begin{array}{c} 1,\;dist\;<\;3\\ 0,\;dist\;\geq \;3 \end{array}\right\}$$
where $D_{i}\;=\;1$ refers to the success of the UAV tracking moving platform at step i, $D_{i}\;=\;0$ refers to tracking failure, and N refers to stopping the tracking test after updating fixed N time steps.

Table 3 shows the RMSE and TSR values of the tracking modules compared with the PID controllers (a complete description of PID controllers and parameters are shown in Appendix A). The moving platform has four different types of motion trajectories, including linear, circular, and random motion. The results show that, although the tracking performance of the PID method is more stable, when encountering complex motion, the RMSE of the proposed method is smaller. Compared with the VIEW percent indicator, the PID method is more likely to cause the loss of the tracking target. Figure 7 shows the trajectory of the UAV in the x-y plane while following a moving platform. Figure 7c,d show that the PID method could fit the motion trajectory of the platform, but it has a large response hysteresis and response error; therefore, compared with the PID method, the proposed method is more effective in solving the problem of tracking large-scale changes in moving platforms.

In order to test the capability of the generalization and robustness of the proposed method, experiments were conducted to analyze the effects of sensor noise on the tracking model. These experiments simulated the measurement noise of the sensors, which limits the ability to acquire the ground-truth motion information of the UAV and moving platforms. We added Gaussian noise (μ = 0 and σ = 1, 2, 3, 4) to observations obtained from the UAV, and the results of the noise influence are shown in Figure 8, and the confidence intervals of the data points are specified in Appendix B. Both models degrade in performance as more noise is applied to the UAV observation. However, the performance of the proposed model does not degrade as quickly as that of the PID algorithms and even performed tracking tasks better than the PID algorithms.

On the other hand, we tested the effects of intermittent measurements on the tracking method. In the process of the UAV tracking the moving platform, the initial state of the system is $S_{0}$, and then the system needs to update the state at a fixed frequency and stop the tracking process after updating the fixed N steps (arriving final state $S_{N}$). Therefore, we set time steps L during which the observations were not updated. Furthermore, the random starting interval $S_{i}$ of [$S_{0}$,$S_{N-L}$] was generated by a random number generation algorithm, namely the linear congruential method [42], and then the intermittent measurements interval [$S_{i}$,$S_{i+L}$) was obtained. This simulated the incomplete information acquired by the sensors—the tracking results for the loss of motion information in an information acquired by the sensors. The tracking results for the loss of motion information in an intermittent measurement environment are shown in Table 4. The results show that, in most cases, the performance of the proposed method is better than that of the PID algorithms.

The results of these three tracking experiments indicate that our algorithms are outperformed by PID controllers in some conditions and are more tolerant of noisy and intermittent measurements than PID method.

### 4.3. Autonomous Landing Tests

Once the tracking methods and landing methods are obtained, they can be used together to control the autonomous landing of the virtual UAV. The moving platform changes its position in the *x*-*y* plane to achieve the four different movement types. The initial position of the UAV is an altitude of 5.0 m with the vertical velocity controller constrained by the *x* and *y* axis errors, and this strategy retains the UAV’s ability to carry out autonomous tracking.

To evaluate the UAV landing method proposed in this paper, we introduced the landing success rate as the evaluation indicator of the landing algorithm. A landing trial is considered successful when the horizontal and vertical distance between the UAV and the moving platform are less than the fixed threshold.

Table 5 shows the success rate based on 100 times landing test episodes, compared with the PID algorithm, the proposed algorithm reduces the success rate by 2~3% when the moving platform performs a linear or circular motion, however, the landing success rate of the proposed algorithm is significantly improved, such as when the moving platform performs random motion. In addition, Figure 9 shows the trajectory of the UAV in the *x-y-z* plane while landing on a moving platform. The success rate and landing trajectory suggest that the PID control could not explore the unknown nonlinearity in its control architecture and thus, usually leads to a suboptimal performance, but the proposed approach succeeded in learning the landing maneuver in complex environments.

Additionally, in order to further validate the performance of our proposed model, we compare the landing success rate of the proposed method with other deep reinforcement learning methods. The results of the test phase are summarized in Figure 10. The bar chart compares the performances of the DQN-single [30], DQN-multi [30], DDPG (two-dimensional) [5,32], and the proposed method. In all tests, the moving platform performed a uniform straight-line motion at a speed of 3 m/s. In this case, the proposed method has the highest success rate with an accuracy of 93%, the DDPG (two-dimensional) has the similar performance (89%). The DQN-single follows with a score of (85%). The DQN-multi score is significantly lower (80%). The results show that the method proposed in this paper can more effectively solve the problem of a UAV’s autonomous landing in high-dimensional and continuous action space.

## 5. Conclusions and Future Work

A hybrid strategy-based autonomous tracking and landing method is proposed. With the help of reinforcement learning and an end-to-end neural network, the proposed method requires no prior information of the moving platform and the UAV can work well in a noisy measurement and intermittent measurements. Compared with the PID methods, the proposed method shows good performance when the platform moves in a complex trajectory. Considering the control strategy of the UAV landing (or other types of robot complex tasks) in unpredictable and harsh environments, other reinforcement learning algorithms can also be applied to our neural network training process, which improves the generality of our control framework. The developed algorithm also has some limitations. For example, due to the complexity and continuity of the state and action space of the UAV landing problem, the *z*-axis is based on heuristic rules and is not included in the solution space of reinforcement learning. Therefore, we would consider including the *z*-axis in the action space in future work. At the same time, we would use the dji-m210 UAV to further verify the robustness and effectiveness of the proposed method, and some offline training controller and online optimizer solutions would be tested.

## Figures and Tables

**Figure 1 sensors-20-05630-f001:**
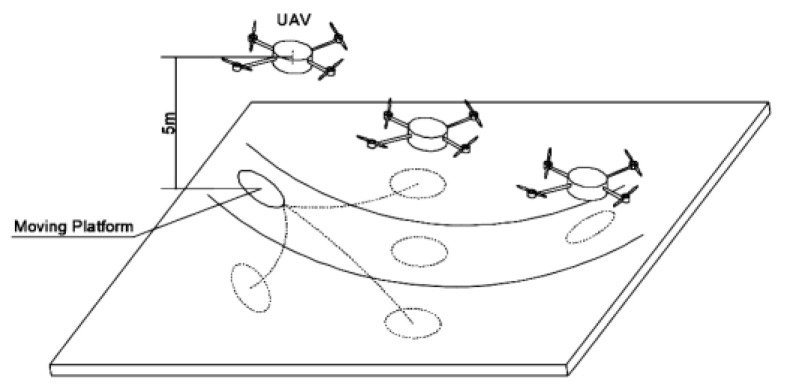
Process of an unmanned aerial vehicle’s (UAV) autonomous tracking and landing, which includes a moving platform and a UAV, in which the UAV’s task is to achieve the tracking of the moving platform and landing on it.

**Figure 2 sensors-20-05630-f002:**
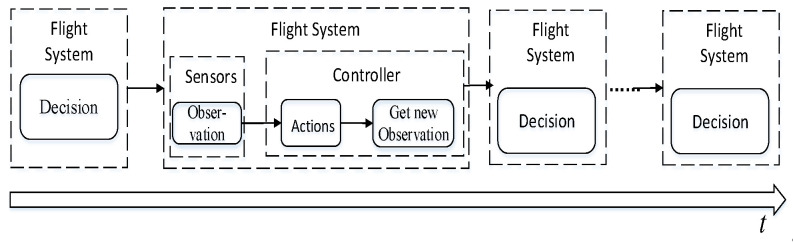
At each moment, the UAV sequential decision process can be divided into three steps: obtaining state information of the current system, performing actions, and updating the state.

**Figure 3 sensors-20-05630-f003:**
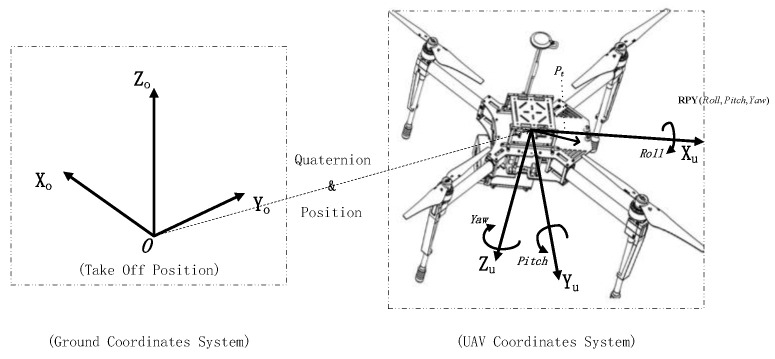
The UAV coordinate system and ground coordinate system are constructed separately, and the pose and velocity of the moving platform and UAV are obtained through coordinates.

**Figure 4 sensors-20-05630-f004:**
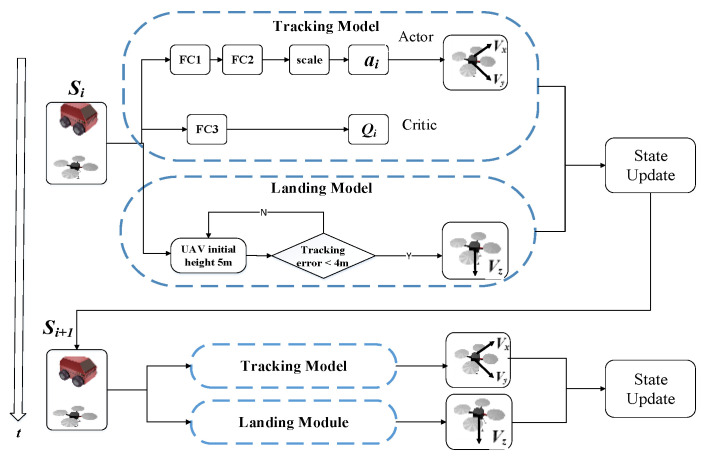
Hybrid strategy consisting of two parts: tracking and landing modules. The tracking module introduces the deep reinforcement learning method, and the landing module adopts heuristic rules, so as to land the UAV on the platform.

**Figure 5 sensors-20-05630-f005:**
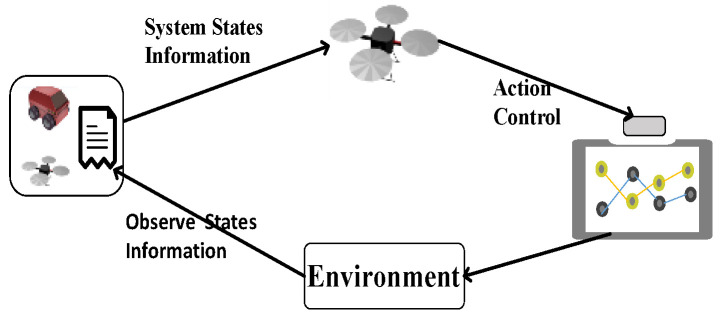
At each step, the UAV observes the environment and interacts with it through actions while receiving immediate signals and updating information of the environment. After multi-step decisions, the UAV acquires decision-making experience and then optimizes the entire task sequence.

**Figure 6 sensors-20-05630-f006:**
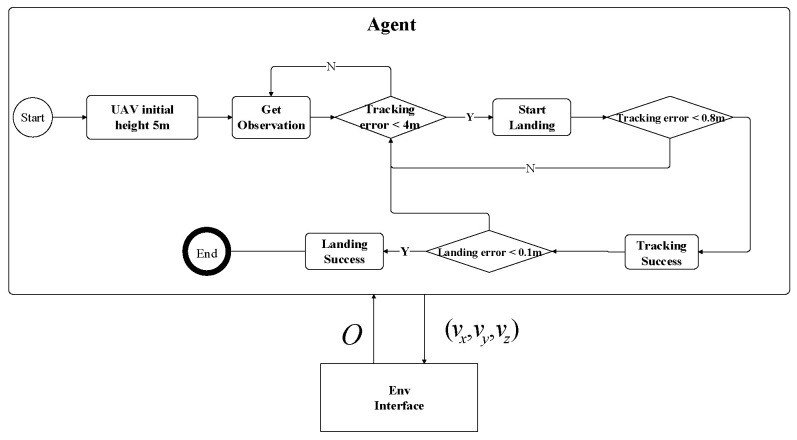
The simulation experiment includes two parts: the tracking and landing experiment. Once the UAV can track the moving target stably, it will adjust the altitude and land on the moving platform.

**Figure 7 sensors-20-05630-f007:**
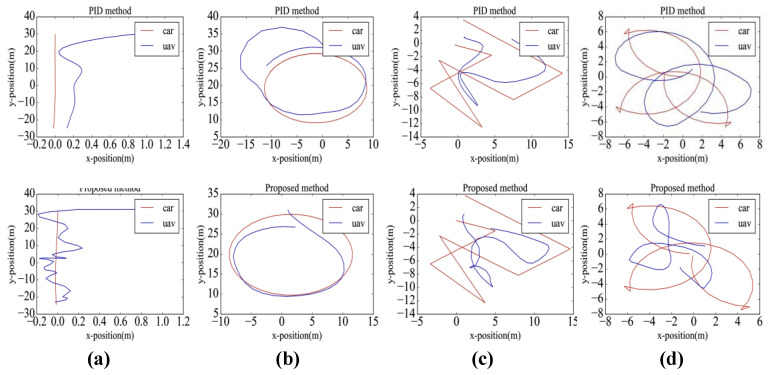
Trajectory of UAV while tracking a moving platform: moving platform for (**a**) linear motion, (**b**) circular motion, (**c**) random motion, and (**d**) random motion.

**Figure 8 sensors-20-05630-f008:**
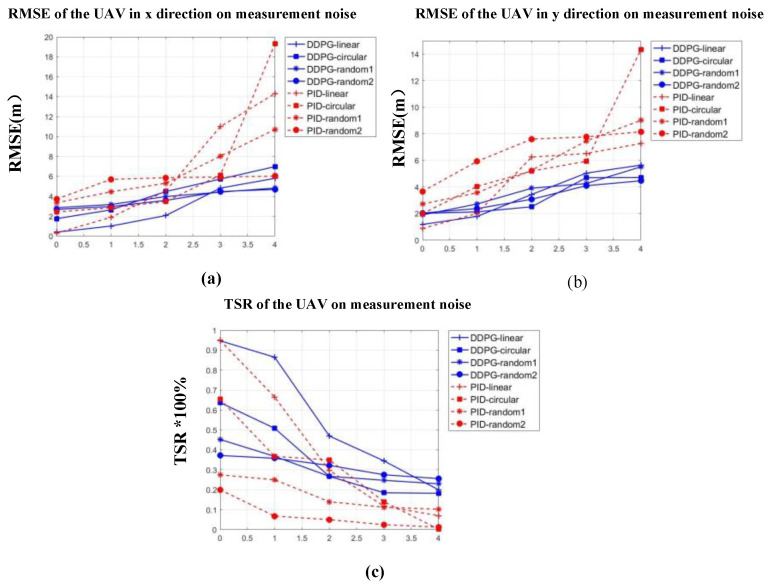
Tracking results on measurements error: (**a**) root-mean-square error (RMSE) of UAV in the x-direction, (**b**) RMSE of UAV in the y-direction, and (**c**) tracking success rate (TSR) of UAV.

**Figure 9 sensors-20-05630-f009:**
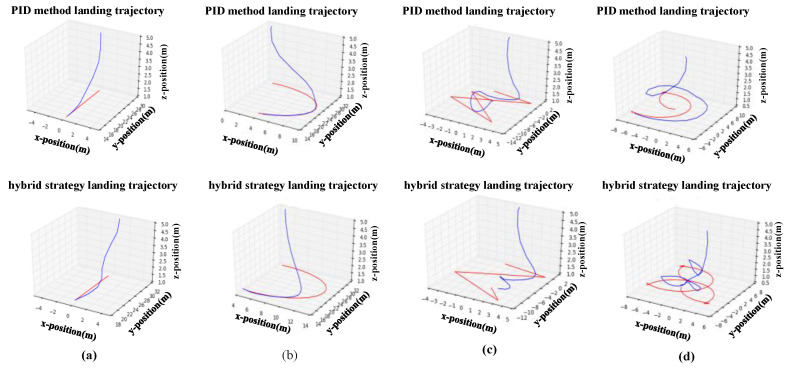
Trajectory of UAV while landing on a moving platform for (**a**) linear motion, (**b**) circular motion, (**c**) random motion, and (**d**) random motion.

**Figure 10 sensors-20-05630-f010:**
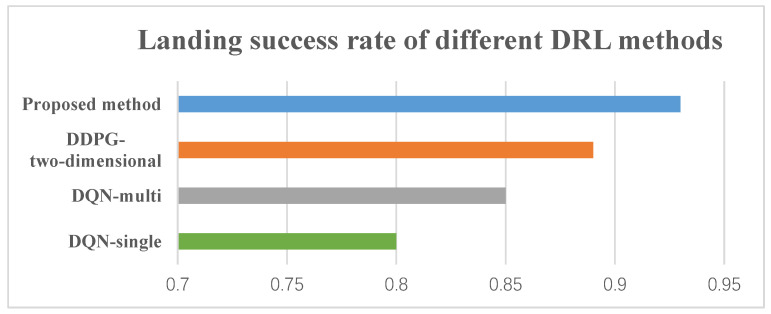
The results of landing simulation with different deep reinforcement learning (DRL) methods. The moving platform performed a uniform straight-line motion at a speed of 3 m/s.

**Table 1 sensors-20-05630-t001:** Network parameters.

Network Name	Number of Hidden Layers	Activation Function
FC1	30	relu
FC2	2	tanh
FC3	30	relu

**Table 2 sensors-20-05630-t002:** The vertical speed generated by heuristic rules.

*Velocity* (*m/s*)		*Dist*		
		0 < *dist* < 0.8	0.8 ≤ *dist* ≤ 4	*dist* > 4
*Height* (*m*)	0 ≤ | *height* | ≤ 0.1	0	0.5 × *height*	−1
	0.1 < | *height* | ≤ 3.5	0.5 × *height*	0.5 × *height*	−1
	|*height*| > 3.5	0.5 × *height*	0.5 × *height*	0

**Table 3 sensors-20-05630-t003:** Root-mean-square error and tracking success rate (TSR) of the unmanned aerial vehicle (UAV) for tracking a moving platform.

Controller	PID Method				Proposed Method				Units
Movement	Linear	Circular	Random1	Random2	Line	Circular	Random1	Random2	
x axis (RMSE)	0.2969	2.4204	3.3479	3.7198	0.3896	1.7466	2.8444	2.6720	m
y axis (RMSE)	0.8809	1.9919	2.7271	3.6738	1.1834	2.0197	1.9053	2.0395	m
TSR (%)	95.0	65.5	27.5	20.3	94.8	63.8	45.3	37.2	--

**Table 4 sensors-20-05630-t004:** Tracking results in an intermittent measurement environment. Root-mean-square error and percentage of steps of the moving platform in the field of view of the UAV are shown in this table.

Controller		PID Method			Proposed Method		
	Values	x axis (m) (RMSE)	y axis (m) (RMSE)	TSR (%)	x axis (m) (RMSE)	y axis (m) (RMSE)	TSR (100%)
Movement							
linear		0.3991	2.4481	78.60	0.3764	2.0297	90.50
circular		6.7256	8.2300	15.50	2.1986	2.5640	59.75
random1		10.3374	10.1190	15.00	3.1644	2.3531	43.00
random2		6.2291	6.0182	12.00	2.6294	3.1471	35.00

**Table 5 sensors-20-05630-t005:** Landing success rate for several movement types.

Movement Type	Linear	Circular	Random1	Random2
PID method	97%	85%	56%	41%
Proposed method	94%	83%	70%	74%

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
