# Peer review of "UAV Autonomous Tracking and Landing Based on Deep Reinforcement Learning Strategy"

_sensors, 2020, doi:10.3390/s20195630_

Round 1
Reviewer 1 Report
To deal with the UAV motion control problem in an unpredictable and harsh environment, this paper presents a UAV autonomous tracking and landing approach based on deep reinforcement learning strategy. Overall, this paper is well-rewritten. It might be accepted if the following issues can be addressed.
- Markov decision process is used. I am confused about the function of the Markov decision process in this paper. It seems some rules for landing have been given in sec. 3.1.
- The contributions of this paper should be explained by comparing it with some studies.
- To apply such a network model training, how to get data for a practical system considering drone safety.
- Section 4: change the title to "Simulation results" because the authors just use a simulation tool to do some tests, not apply to a real quadrotor.
- Future research directions should be pointed out in conclusion part.
Reviewer 2 Report
Reviewer Comments for Author
Comments to Author:
General Comments:
I made efforts to review the manuscript carefully and found that authors have done an excellent job. Here I am giving brief remarks on the current form of a paper.
Xie et al. have studied the partially observable Markov decision process (POMDP) mathematical model to deal with the UAV motion control problem in the unknown and harsh environments. Further, this approach was calibrated/validated and compared against the PID method. The proposed model has the potential to improve UAV Autonomous Tracking. Thus, it is an encouraging study that presents the comparison of three different methods for removing the background signal from the UAV acquired images. The authors should discuss the importance of the results and discussion section to be included in the main body of the manuscript. The conclusion of this paper is too short I advise adding few lines related to its application and limitation of the developed model.
Please see the comments and suggestions included in the following paragraph.
Abstract
I suggest the authors improving the abstract though it’s very well written but there is scope to improve for better clarity.
Statistical analysis results are not shown for the different methods. I suggest the authors mentioning the errors for the proposed and PID methods. Also mention the implications of the findings.
Introduction
The introduction part is very well presented. The authors have carefully cited the work of the most recent research done in this area.
Problem Definition
This section is greatly presented and comprises details of the mathematical model.
Experimental Results
This section is very well written. Results are very well presented and supported by the neat and clean plots with the statistical matrices.
The authors should present a flow chart based methodology for a better understanding of the proposed simulation. Also, authors may perform the model sensitivity analysis and include it in this section.
Conclusions and Future Work
This section is very well written. The findings of the present study are summarized in a logical and organized manner. The conclusion of this paper is too short I advise adding few lines related to its application and limitation of the developed model.
Overall, the authors have used a good standard of English language and presented all the sections and subsections in a logical and organized manner. This study cannot be published in the present form due to minor comments. The modeling results of this study is proving the model with a good performance against the PID method. Therefore, I recommend revising the manuscript based on my comments.
Reviewer 3 Report
Comments to:
UAV Autonomous Tracking and Landing Based on DRL Strategy
The paper presents an interesting approach for Autonomous Tracking and landing on a mobile platform based on Deep Reinforcement Learning.
The authors implement a NN that gets the states of the UAV and landing platform and then outputs the control signal (longitudinal and lateral commands) to the UAV. The paper is very interesting and tries to solve a problem that is challenging to the scientific community. However there was an important competition in this regard (MBZirc 2020) and none of the papers of the winning teams have been cited. Authors should consider to add the competition to the introduction and to add citations. For example:
- Baca, T, Stepan, P, Spurny, V, et al. Autonomous landing on a moving vehicle with an unmanned aerial vehicle. J Field Robotics. 2019; 36: 874– 891. https://doi.org/10.1002/rob.21858
- M. Beul, S. Houben, M. Nieuwenhuisen and S. Behnke, "Fast autonomous landing on a moving target at MBZIRC," 2017 European Conference on Mobile Robots (ECMR), Paris, 2017, pp. 1-6, doi: 10.1109/ECMR.2017.8098669.
I have some question to the learning process. You only learn the tracking module, not the landing one. That means that the vertical speed is generated by a deterministic formula. Could you represent the height as a function of the distance to the landing platform? Also, when learning the tracking module, are the commands generated at random at first, or else where do you find the initial inputs?
The authors only present simulation experiment and even then, the success rate is not very high for random scenarios (on the 70% in the proposed methods). In addition, the PID controller seems to be not very aggresive, when compared to the proposed ones. That would explain the bad numbers regarding to visibility of the platform of the classical controller in Table 2. What method have the authors followed in order to adjust the PID parameters?
Another interesting study is the performance of the methods when noise is added to the observations. The degradation in the PID method is usually due to the derivative term and the noise observations are usually filtered by using a bayesian filter (for example Kalman filtering). Have you tried it? Also, there is an abnormal situation regarding to view percentage in random2 scenario: the view perc. raises when the noise is added. Could you explain it?
There are some comments considering the language style that I found not easily understandable.
Please rephrase: "while tracking stability and robustness are sensitivity to parameter selections"
Line 46: "by applying SDC-based" the SDC acronym has not been introduced
"However, development of accurate prediction models requires a tiresome control design effort." ?? How a inferring a model has to do with control design effort? Please rephrase
Line 53: "the main issue with sliding mode control is chattering and high control demand" the authors should include a citation to give light to such information.
Line 56: "Also fuzzy logic controllers can be combined with PID to model the system in a more realistic manner" I would not say that fuzzy controllers model the system in a more realistic way, but rather it can improve the PID performance by changing its parameters on some fuzzy rules. But to me, this is very different perform a realistic model of the system (which should be done in MPC).
Line 83: "perception and relative pose estimation or control trajectory" or = and?
Line 173: "Differs from previous work [39]," Please rephrase. Also, it seems that the reference is not what it was supposed to be, was it [21]?
Line 251: 8 Gb DDR4, did you mean 8 GB? (B = bytes, b = bit)
Reviewer 4 Report
This paper presents Deep Deterministic Policy Gradients algorithm to control the landing maneuver of a UAV on a moving landing platform. The methodology is interesting, presents innovative aspects in comparison with the related work, and the presented results compare favorably with the PID method.
The text and figures require substantial improvement, including the following:
- Acronyms must be introduced the first time they appear (e.g., SDC, DRL);
- There are some grammar mistakes, or badly structured sentences;
- Some equation formatting has problems (e.g., Eq. 5, but there are others);
- Variables in the text should be in italics, which not always happens;
- Footnote 1 appears in the page following its appearance in the text;
- The quality of the figures with results is mostly bad, namely Fig. 6 and Fig. 7.
Besides these editing aspects, I have the following remarks:
- Eq. (3): It is not explained why the authors decided to include a reward term for 0.8<=dist<=6. Its impact should be explained.
- Section 3.1, Tracking Module: the network structure, namely the use of two layers FC1 and FC2 should be better explained in the text.
- Section 4.2: "VIEW percent" does not seem to be a good designation for this metric.
- Section 4.2: "On the other hand, we tested the effects of intermittent measurements on the tracking methods" - the random intermittence pattern should be clearly described.
- Fig. 7: Confidence intervals for the obtained points should be provided, or at least commented in the text.
Round 2
Reviewer 1 Report
No further comments.